# A novel method for *in silico* assessment of Methionine oxidation risk in monoclonal antibodies: Improvement over the 2-shell model

**Davide Tavella**[1]*, **David R. Ouellette**[1], **Raffaella Garofalo**[2], **Kai Zhu**[3], **Jianwen Xu**[1], **Eliud O. Oloo**[3], **Christopher Negron**[1]*, **Peter M. Ihnat**[4]

1 AbbVie Bioresearch Center, Worcester, Massachusetts, United States of America, 2 AbbVie Deutschland GmbH & Co. KG, Analytical Innovation and Mass Spectrometry, Knollstrasse, Ludwigshafen, Germany, 3 Schrödinger, Inc., New York, New York, United States of America, 4 Regeneron Pharmaceuticals Inc., Tarrytown, New York, United States of America

* davide.tavella@abbvie.com (DT); christopher.negron@abbvie.com (CN)

**Data Availability Statement:** All relevant data are within the paper and its Supporting Information files. Some data regarding the molecules provided by AbbVie cannot be shared publicly because of

## Abstract

Over the past decade, therapeutic monoclonal antibodies (mAbs) have established their role as valuable agents in the treatment of various diseases ranging from cancers to infectious, cardiovascular and autoimmune diseases. Reactive groups of the amino acids within these proteins make them susceptible to many kinds of chemical modifications during manufacturing, storage and *in vivo* circulation. Among these reactions, the oxidation of methionine residues to their sulfoxide form is a commonly observed chemical modification in mAbs. When the oxidized methionine is in the complementarity-determining region (CDR), this modification can affect antigen binding and thus abrogate biological activity. For these reasons, it is essential to identify oxidation liabilities during the antibody discovery and development phases. Here, we present an *in silico* method, based on protein modeling and molecular dynamics simulations, to predict the oxidation-liable residues in the variable region of therapeutic antibodies. Previous studies have used the 2-shell water coordination number descriptor (WCN) to identify methionine residues susceptible to oxidation. Although the WCN descriptor successfully predicted oxidation liabilities when the residue was solvent exposed, the method was much less accurate for partially buried methionine residues. Consequently, we introduce a new descriptor, WCN-OH, that improves the accuracy of prediction of methionine oxidation susceptibility by extending the theoretical framework of the water coordination number to incorporate the effects of polar amino acids side chains in close proximity to the methionine of interest.

## Introduction

Since the first approval of a monoclonal antibody (mAb) for therapeutic use in 1986 [1], the development of mAb-based therapeutics has gained increasing interest in the pharmaceutical

their proprietary nature. In this work, we have used two experimental datasets to validate the in silico predictions of methionine oxidation propensities and to reach the conclusions drawn in the manuscript. The first dataset, referred to in the manuscript as "clinical stage therapeutic (CST) antibodies dataset", was derived from existing data, which are openly available at https://doi.org/10.1080/19420862.2017.1290753. This is the largest dataset used in this work, counting 14 antibodies and a total of 46 methionines. The second dataset, referred to in the manuscript as "internal dataset", is derived from antibodies and ADCs currently in development at AbbVie. Therefore, due to its proprietary nature, supporting data cannot be made openly available. This is the smallest dataset used in this work, counting 7 antibodies and 2 ADCs for a total of 26 methionines. We want to highlight that the conclusions drawn from both datasets are in total agreement and the "internal dataset" serves as a further validation of the methods described in the manuscript. Moreover, in the supporting information S2 Table and S3 Table, we have disclosed more details regarding the calculations derived from the molecular dynamics simulations for both datasets.

**Funding:** AbbVie sponsored and funded the study, contributed to the design, participated in the collection, analysis, and interpretation of data, and in writing, reviewing, and approval of the final publication. Davide Tavella, Christopher Negron, David R. Ouellette, Raffaella Garofalo and Jianwen Xu are employees of AbbVie and may own AbbVie stock. Peter M. Ihnat was an employee of AbbVie at the time of this study. Desmond, Prime, Maestro and BioLuminate are products sold by Schrödinger, Inc. Schrödinger, Inc. provided support in the form of salary for authors Eliud O. Oloo and Kai Zhu but did not have any additional role in the study design, data collection and analysis, decision to publish or preparation of the manuscript. The specific roles of these authors are articulated in the "author contributions".

**Competing interests:** The authors have declared that no competing interests exist.

industry. To date, more than 70 mAbs have been approved and many more are currently in discovery and clinical phases [2].

A mAb consists of two fragment antigen-binding (Fab) regions and one fragment crystallizable (Fc) region. The Fab fragments contain the variable regions (Fv) that are responsible for antigen-binding specificity through the complementarity-determining region (CDR). The Fc fragment contains the constant regions and is responsible for the mAb function, via interactions with Fc receptors, and *in vivo* disposition, via interactions with the neonatal Fc receptor (FcRn), which may extend serum half-life [3,4].

During production and purification of a therapeutic mAb and manufacture of clinical supplies, the candidate molecule must go through a series of potentially stressful unit operations (e.g. antibody generation in culture media, purification, formulation, and storage) before dosing in patients [5,6]. Given the intrinsic flexibility and the dynamic nature of the antibody structure in solution, and the presence of many functional groups in the amino acids side chains, the occurrence of chemical modifications is much higher for mAbs than for small molecule drugs. Since these modifications may negatively impact the intended biological functions of the antibody, it is efficient and cost effective to predict the chemical modification propensity of potentially liable sites as early as possible during the development process [7–10]. Among the 20 natural amino acids in mAbs methionine, cysteine, tryptophan, tyrosine and histidine residues are theoretically susceptible to oxidation. The sulfur in methionine, present as a thioether (R-S-R′), has a low oxidation potential and therefore a large number of oxidizing species can oxidize this residue [11]. The oxidation reaction of methionine can occur via two distinct mechanisms, depending on the oxidant species: oxidants such as HOCl, $H_2O_2$, and singlet oxygen directly oxidize methionine (Met) to methionine sulfoxide form (MetO) via a formal oxygen transfer by a two-electron oxidation; radicals such as HO• and metal ions such as $Fe^{III}$ and $Cu^{II}$, in contrast, oxidize methionine in a one-electron oxidation [11].

The oxidation of methionine to the methionine sulfoxide form has been widely reported for many types of proteins [12–15]. Although methionine oxidation has been reported to be reversible *in vivo* through the activity of the methionine sulfoxide reductase (MsR), which catalyzes the thioredoxin-dependent reduction of MetO back to Met, it is not yet clear whether, in the context of therapeutic mAbs, an oxidized methionine can be readily reduced back to Met [11,16–18]. In light of these observations, in order to develop an effective and safe therapeutic mAb, it is highly desirable to predict and understand the role of potential chemical liabilities, such as methionine oxidation sites, during molecular profiling that occurs in late discovery or early development of the candidate. Oxidation of Met residues in mAbs has been reported as a result of exposure to oxidizing agents [19], photo-irradiation [20], or simply during storage [21]. Depending on the site of the modification, this oxidation has been shown to alter a mAb's conformational structure [22], stability [23], binding to Protein A and Protein G [24,25], loss of antigen binding [26], reduced binding to FcRn [27] and shorter *in vivo* half-life [28]. Given the potential impact of methionine oxidation on drug development, early identification of this liability provides the opportunity to reengineer the susceptible site during the late discovery or early pre-clinical development phases. Conversely, if the mAb candidate with an oxidizable methionine cannot be reengineered and is advanced to the development phase, early identification allows for timely establishment of a risk management strategy of this critical quality attribute (CQA). The *in silico* prediction of potential liabilities is an essential component of developability screening that also relies on a variety of stress conditions and extensive analytical characterization to provide a developability risk assessment of the lead candidate. Therefore, there is a significant interest in developing new descriptive variables that may be used with current modeling software to provide *in silico* prediction of chemical liabilities in the antibody sequence.

Several advancements have been reported previously that predict methionine oxidation in proteins. Based on the observation that methionine residues on the surface of a protein are oxidized at a higher rate than the buried ones [29–34], it has been hypothesized that the oxidation reaction is largely governed by the solvent exposure of the methionine side chain. For this reason, the solvent-accessible surface area (SASA) of the methionine side chain from a predicted antibody structure is a commonly used parameter to predict oxidation propensity of methionine residues in mAbs [35,36]. However, Chu et al. showed that SASA fails to explain the oxidation rates of partially buried methionine residues in granulocyte colony-stimulating factor (G-CSF) and α-1 antitrypsin [37]. Specifically, the authors showed that if 2–3 water molecules are present around buried methionine sites, the reaction barrier is similar to that of free methionine. The solvent, in fact, might still access spatially restricted residues via thermal fluctuations. Thus, Chu et al. proposed a "water-mediated" mechanism and developed the 2-shell water coordination number (WCN) as a parameter. Interestingly, WCN correlates better with experimentally measured oxidation rates [37,38]. More recently, Aledo et al. reported a predictive method for methionine residues oxidized *in vivo* in response to oxidative signals based on published proteomic data [39]. Their study identified the three most relevant features that contribute to methionine oxidation: (i) the solvent accessible area of the methionine residue, (ii) the number of residues between the analyzed methionine and the next methionine found towards the N-terminus and (iii) the spatial distance between the sulfur atom in the methionine and the closest aromatic residue [39]. Lastly, Sankar et al. developed a quantitative and highly predictive *in silico* methionine oxidation model for screening early candidates using machine learning and features calculated from the primary sequence, from the mAb structure obtained by homology modeling, and from coarse-grained elastic network models [40].

Despite the success of the models described above, they remain inconsistent with oxidation data regarding partially buried methionines. For instance, Yang et al. observed a good correlation between the solvent-accessible surface area (SASA) of the side chain of methionine residues and the measured oxidation events in the corresponding segments of the mAb [36]. An exception to this observation, however, was represented by a subset of the 121 clinical stage mAbs containing a buried methionine residue in the H3 loop of the CDR (SASA <11%, Fig 1). Yang et al., found that for the 22 antibodies that contain such a feature, the factors affecting the oxidation of methionine at this position are not entirely captured by the solvent accessibility of the methionine side chain. In this study, we present our efforts to develop a more accurate model for methionine oxidation, regardless of location within the protein, using both publicly available and experimentally generated data.

## Materials and methods

### Homology modeling and molecular dynamics simulations

This study considered 7 mAbs and 2 antibody-drug conjugates (ADCs) provided by AbbVie as well as 14 mAbs advanced to clinical stages by other companies. The 14 clinical stage antibodies were: Abituzumab, Dinutuximab, Duligotuzumab, Eldelumab, Fletikumab, Golimumab, Imgatuzumab, Lintuzumab, Lirilumab, Natalizumab, Ofatumumab, Tocilizumab, Tovetumab and Vesencumab. The three-dimensional structures of the variable fragment (Fv) of all antibodies in the study were modeled using the automated protocol implemented in the BioLuminate package, Schrödinger suite version 2019–2 [41–43] (Schrödinger, LLC, New York, NY). Briefly, the python script build_antibody.py, provided within the BioLuminate package, was used with default options to generate the homology models of the Fv regions of the antibodies of interest. Additionally, 10 models of the Fv region of Vesencumab were obtained with the "advanced loop model" option within the antibody prediction tool in

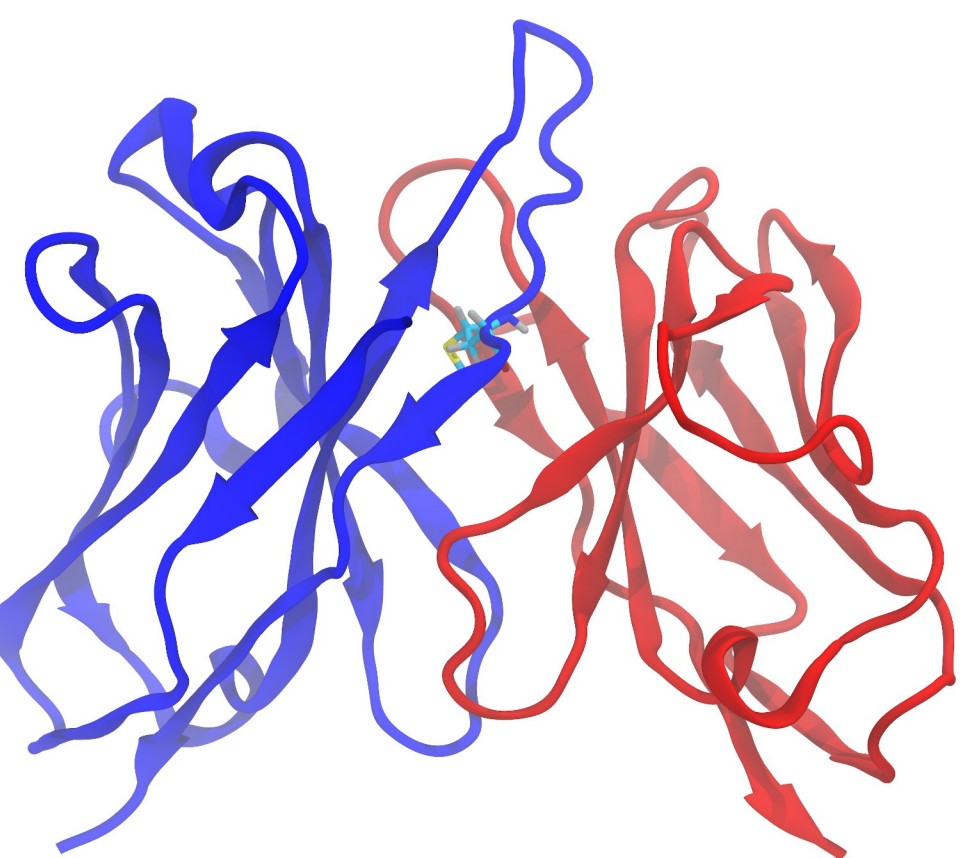

**Fig 1. Three-dimensional structure (homology model) of the Fv region of the antibody Lirilumab.** The heavy chain and the light chain are represented in blue and red, respectively. Methionine 100F, located at the end of the H3 loop, is shown with carbon atoms colored in cyan, oxygen in red, hydrogen in gray, nitrogen in blue and sulfur in yellow.

Maestro (Schrödinger suite version 2019–2), employing *ab initio* structure prediction with Prime [44,45]. Sampling of the methionine side chain conformations in each of the protein structures was achieved using a molecular dynamics (MD) approach. The Fv models described above were solvated in an orthorhombic box with SPC waters and MD simulations were carried out using DESMOND [46], Schrödinger suite version 2019–2 (Schrödinger LLC, New York, NY), in explicit solvent with periodic boundary conditions. The OPLS3 force field [47] was used for each protein. The system was initially relaxed with restraints on solute allowing waters to freely equilibrate, followed by extensive simulation of the entire system without any restraints. Production trajectories were 5 ns long, collected with a 50-ps recording interval at 300K using a standard protocol. The same MD simulation protocol was applied to the crystal structures of the Fab regions of Abituzumab (PDB ID: 4O02), Ofatuzumab (PDB ID: 3GIZ) and Vesencumab (PDB ID: 2QQN). MD simulations were performed at the iForge National Center for Supercomputing Application (NCSA), University of Illinois at Urbana-Champaign on nodes with Nvidia V100 GPUs.

## Analysis of trajectories

The analysis of the structural properties of the three-dimensional models of the Fv region of the 23 antibodies (7 mAbs and 2 ADCs provided by AbbVie, 14 clinical stage antibodies) and the trajectories described in the section above was performed with python scripts using the

Schrödinger Python API. Specifically, the solvent-accessible surface area (SASA) of the side chain of each methionine residue in the Fv region was computed (static SASA or sSASA); the time-averaged value of the SASA of each methionine side chain was calculated from the MD trajectories (dynamic SASA or dSASA); the 2-shell water coordination number (WCN), defined as the average number of water molecules within a radius of 6 Å from the sulfur atom in the side chain, was computed for each methionine residue in the Fv region of the 23 antibodies; similarly, the average number of hydroxyl groups (#OH) in the side chain of tyrosine, threonine and serine residues within a radius of 6 Å from the sulfur atom of each methionine was calculated from the trajectories. SASAs are reported as relative values, the maximum allowed area being described by Tien et al [48].

Representations of the protein structures were produced with VMD (Visual Molecular Dynamics) [49,50] and Tachyon [51].

## Oxidation stress

Oxidized samples of mAb1-7 were generated by diluting to 2 mg/ml in Phosphate buffered saline (PBS), pH 7.4. Samples were subsequently subjected to tert-butyl hydroperoxide (tBHP, Alfa Aesar, Haverhill, Massachusetts) at a final concentration of 0.1% for an incubation time of 24 hours at 21˚C in the dark. The oxidation reaction was subsequently quenched by addition of free methionine in solution at 10 mM final concentration. Samples were stored at -80˚C before further treatment or analysis.

Details for the preparation and analysis of ADC1-2 and for replicates of mAb2, mAb4, mAb7 (S1 Table) are provided in Supporting Information.

It is worth noting that the oxidation condition used to generate the data described here differs from the one reported for the dataset by Yang et al. [36] due to the nature of the oxidizing agent. Both hydrogen peroxide and tBHP oxidize methionine residues by a nucleophilic substitution reaction [52]. $H_2O_2$ can react with oxidation vulnerable amino acids, especially methionines, to generate irreversible modifications such as methionine sulfoxides; tBHP, a tertiary-butyl analog of $H_2O_2$, oxidizes predominantly surface exposed methionine residues, and therefore is used to probe the effect of oxidation of exposed methionines on protein structure and stability [52]. It has been reported, however, that especially for methionines in the Fab of an antibody, forced oxidation studies carried out by $H_2O_2$ or tBHP produce similar oxidation levels and lead to the identification of the same methionine as oxidation prone [19].

## Peptide mapping and oxidation quantification

Each oxidized sample was diluted to 0.5 mg/ml in denaturing buffer, 8 M Guanidine-HCl (GuHCl), pH 8.5 (VWR, Radnor, Pennsylvania), to a final concentration of 6 M GuHCl and reduced in the presence of 5 mM Dithiothreitol for 20 min at 37˚C. After alkylation in 17 mM Iodoacetic acid (Alfa Aesar, Haverhill, Massachusetts) for 20 min at 37˚C samples were buffer exchanged to digestion buffer (10 mM TRIS pH 8) using 0.5 mL Zeba spin desalting columns, 7K MWCO. Exchanged samples were digested for 2 hours at 37˚C using 1:10 enzyme:protein Trypsin/LysC mix (Promega, Madison, USA). After incubation, samples were acidified using 5% TFA (Sigma Aldrich, St. Louis, Missouri) to quench proteolysis reaction and frozen for further analysis.

Peptides generated during proteolysis were separated and analyzed on a Q Exactive™ Plus Mass Spectrometer (Thermo Fisher Scientific, Waltham, Massachusetts) using an ACQUITY Peptide BEH 300 C18 column (300 Å pore size, 1.7 μm particle size, 2.1 mm diameter and 150 mm length, Waters, Milford, MA, USA). The solvents used for chromatographic separation

were 0.1% formic acid in MS grade water (mobile phase A) and 0.1% formic acid in acetonitrile (mobile phase B).

Eluted peptides were sprayed by a HESI-source with a spray voltage set at 3.5 kV, capillary temperature 300˚C, aux gas heater temperature 430˚C, sheath gas and auxiliary gas flow rates of 50 and 15, respectively. Full MS scan was set at microscan 1, resolution 70000, ACG target 3e6, maximum IT 50 ms and scan range 200 to 2000 m/z.

The dd-MS2 was set at microscan 1, resolution of 17500, AGC target 1e5 and maximum IT 150 ms and an isolation window of 2.0 m/z in a top-5 method. Quantification of the methionine oxidation was performed with Thermo Scientific Biopharma Finder 3.1.

## Results

### Prediction of methionine oxidation propensity in clinical stage therapeutic (CST) antibodies: Homology modeling

Availability of reliable and accurate experimental data is a crucial step for the development and validation of any predictive algorithm. At first, we considered the results presented by Yang et al. [36] in their study of methionine oxidation in monoclonal antibodies in the context of forced oxidation by hydrogen peroxide.

Briefly, Yang et al. employed a high-throughput liquid chromatography-mass spectrometry-based method to identify oxidation events in three distinct segments of an antibody resulting from enzymatic cleavage: light chain, Fab portion of heavy chain, and Fc. The method was applied to 121 clinical stage mAbs and for each segment of these molecules the fraction of the native (non-oxidized) species was reported together with the fraction of oxidized products. This approach led to ambiguous assignment of the oxidation events to a specific residue for segments containing more than one methionine.

To better evaluate the accuracy of the predictions based on these structural features, among the 22 mAbs identified by Yang et al. we selected the 14 that displayed either 0% or 100% of non-oxidized species. Cases that are partially oxidized are omitted since it is impossible to rule out that the measured oxidation signal for the segment results from only one methionine or from the combination of low-level oxidation from multiple methionine residues.

In order to develop a more accurate predictive method for methionine oxidation propensity in monoclonal antibodies, we built the three-dimensional structures of the Fv of these 14 mAbs containing a buried methionine in the CDR-H3 loop using homology modeling. Based on previously described predictive methods [35,38], from these structures we calculated the static SASA (sSASA) for methionine side chains and, using MD simulations, the time-averaged SASA (dSASA) and water coordination number for methionine side chains (S2 Table). We then compared the calculated features with the experimental data described by Yang et al.

Finally, we used the calculated structural features described above to predict the oxidation propensity of each methionine in the Fv portion of the heavy chain for the 14 mAbs. We defined threshold values for each feature to obtain a binary descriptor, with values of 0 and 1 corresponding to non-oxidation prone and oxidation prone methionine residues, respectively. This binary descriptor allowed for counting the oxidation events in the Fab portion of heavy chain (Table 1). In the case of the sSASA and dSASA, we labeled a methionine as oxidation prone when its relative SASA is greater than 15%. Previous studies report similar cut-off thresholds, 8–11%, with variability due to the use of different values of the maximum possible SASA for the methionine side chain [36]. For the water coordination number (WCN), we defined a methionine to be oxidation prone if at least 6 water molecules are within 6 Å from the sulfur atom. These values were chosen to maximize the agreement between the predicted values and the experimental results in this specific dataset.

**Table 1. Comparison of experimental results and in silico prediction of oxidation events for the 14 CST antibodies.**

| Name | Number of Fd methionine | Observed Ox event (s) | Predicted Ox event (s) ADIMAB | Predicted Ox event (s) sSASA | Predicted Ox event (s) dSASA | Predicted Ox event (s) WCN | Predicted Ox event (s) WCN-OH |
|---|---|---|---|---|---|---|---|
| Abituzumab | 4 | 0 | 0 | 0 | 0 | 0 | 0 |
| Dinutuximab | 4 | 2 | 2 | 2 | 1 | 2 | 2 |
| Duligotuzumab | 2 | 0 | 0 | 0 | 0 | 0 | 0 |
| Eldelumab | 5 | 2 | 2 | 2 | 2 | 2 | 2 |
| Fletikumab | 3 | 0 | 0 | 0 | 0 | 0 | 0 |
| Golimumab | 4 | 0 | 0 | 0 | 0 | 0 | 0 |
| Imgatuzumab | 3 | 1 | 0 | 0 | 0 | 0 | 0 |
| Lintuzumab | 3 | 0 | 0 | 0 | 0 | 0 | 0 |
| Lirilumab | 3 | 1 | 0 | 0 | 0 | 0 | 1 |
| Natalizumab | 3 | 0 | 0 | 0 | 0 | 0 | 0 |
| Ofatumumab | 3 | 0 | 0 | 0 | 0 | 0 | 0 |
| Tocilizumab | 2 | 1 | 0 | 0 | 0 | 0 | 1 |
| Tovetumab | 3 | 0 | 0 | 0 | 0 | 0 | 0 |
| Vesencumab | 4 | 2 | 1 | 1 | 1 | 1 | 2 |

For each mAb, we counted the total number of oxidation prone methionine residues, comparing the results with the experimental observations. We found that the predictions based on sSASA, dSASA and WCN provided similar results for the 14 mAbs in the dataset, with values of sensitivity (true positive rate, TPR) and specificity (true negative rate, TNR) within one standard deviation. These methods correctly predict all the negative oxidation events. However, they performed poorly on prediction of positive oxidation events on this dataset, with a TPR of about 0.5.

To improve upon the aforementioned methods, we expanded the concept of water coordination number in the context of the theoretical framework developed by Chu et al. [38]. In their work, Chu et al. computationally characterized the oxidation reaction of the sulfur atom in the methionine side chain by hydrogen peroxide. The authors identified the limiting step in the oxidation reaction as the charge separation that occurs between the two oxygen atoms in the hydrogen peroxide molecule. In the model, this charge separation is favored by the network of hydrogen bonds that is formed between the hydrogen peroxide and the surrounding water molecules. As noted by the authors, however, the polar side chains of other amino acids surrounding the methionine could form such hydrogen bonds when no water molecules, or few, are available.

In light of these observations, we introduce here a new parameter, WCN-OH, that takes into account both water molecules and polar side chains containing a hydroxyl group (threonine, serine and tyrosine) within 6 Å from the sulfur atom in the methionine (Fig 2). This parameter is calculated from MD trajectories computing at each timestep and averaging both the WCN and the number of hydroxyl groups (#OH) in neighboring side chains. The number of hydroxyl groups showed better agreement with the experimental results than other hydrogen bond donors/acceptors (i.e. amide groups). This finding can be explained with the difference in electronegativity between oxygen and nitrogen and the resulting different partial charges in the hydroxyl and amide groups.

The parameter WCN-OH is used to label oxidation prone methionine residues when one of the following conditions is satisfied:

1. WCN is greater than 6, or

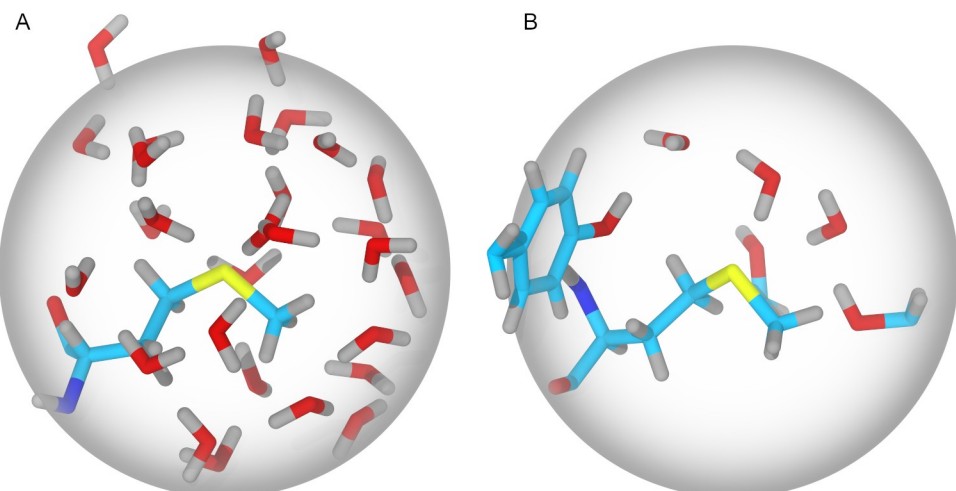

**Fig 2. Representation of the two-shell water coordination number (WCN) and the WCN-OH variant.** (A) For a methionine with the side chain exposed to the solvent, the water molecules within 6 Å from the sulfur atom are shown. (B) For a methionine with the side chain only partially exposed to the solvent, water molecules and side chains containing a hydroxyl group are shown. Carbon atoms are colored in cyan, oxygen in red, hydrogen in gray, nitrogen in blue and sulfur in yellow. The sphere represents the 2-solvation shell (radius 6 Å).

2. WCN is greater than 0.1 and #OH is greater than 1.5.

Condition 1 is equivalent to the WCN method described above; condition 2 is based on the rationale that the methionine side chain must be accessible to a transient water molecule and that more than one hydroxyl group from neighboring side chains have to be present within 6 Å from the sulfur atom. As for the other methods, threshold values were chosen to maximize the agreement between the predicted values and the experimental results in this dataset.

With the threshold values described in condition 1 and 2, the prediction method based on this new parameter provided good agreement with the experimental results (Table 1) and a significant improvement in the prediction sensitivity (Fig 3) compared to the other methods described above. With this new method, only one oxidation event out of the 46 events is wrongly assigned and therefore the resulting sensitivity is 0.88±0.05.

### Prediction of methionine oxidation propensity in CST antibodies: High-resolution experimental structures and advanced H3 loop modeling

In the previous section, we showed that both the WCN and the exposed surface of the side chain can be used to predict the oxidation propensity of methionine residues in mAbs. Unfortunately, these features were not able to account for all the factors that can lead to the oxidation reaction. As a result, we surmised that the poor correlation between WCN, sSASA and dSASA and experimentally determined methionine oxidation may be the consequence of estimating the three-dimensional structures of the mAbs using homology modeling.

To exclude this hypothesis, we considered three CST antibodies among the 22 with at least one methionine in the CDR-H3 loop for which there is an X-ray crystal structure available in the RCSB Protein Data Bank: Abituzumab (PDB ID: 4O02), Ofatuzumab (PDB ID: 3GIZ) and Vesencumab (PDB ID: 2QQN). For these structures we calculated the sSASA of the eleven methionine residues in the heavy chain of the Fab fragment and, using molecular dynamics simulations, the time-averaged values of the water coordination number and dSASA. Additionally, we computed the WCN-OH parameter as described in the previous section.

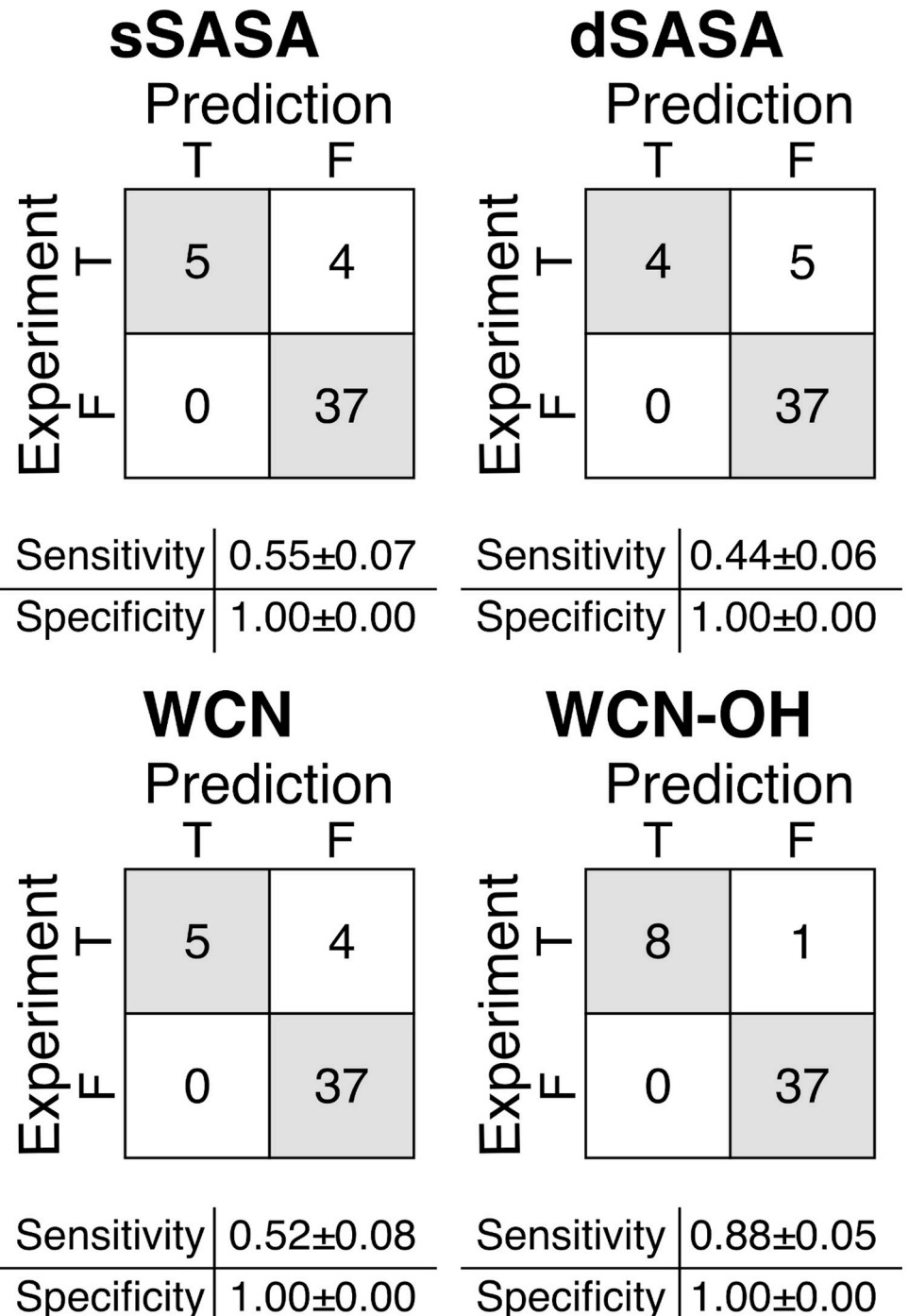

**Fig 3. Confusion matrices for the in silico prediction of the oxidation events of the 14 CST antibodies.** Confusion matrices reports TP, FN on the first row and FP, TN on the second row. Sensitivity calculated as TP/(TP+FN); Specificity calculated as TN/(TN+FP). TP = True Positive, TN = True Negative, FP = False Positive, FN = False Negative. Error for the sensitivity and specificity were estimated from 50 bootstrap replicates.

The results, shown in Table 2, illustrate that the methods based on sSASA, dSASA and WCN only correctly predict the experimentally observed oxidation of one methionine residue, Vesencumab H:100B, but these methods are unable to predict the second oxidation event for

**Table 2. Comparison of in silico prediction of oxidation events based on homology models and crystal structures.**

| Name | Methionine | sSASA (Å²) | | dSASA (Å²) | | WCN | | WCN-OH (#OH) | |
|---|---|---|---|---|---|---|---|---|---|
| | | Model | Crystal structure | Model | Crystal structure | Model | Crystal structure | Model | Crystal structure |
| Abituzumab | H:34 | 3.26 | 0.00 | 1.89 | 0.44 | 1.05 | 0.67 | 0.71 | 0.42 |
| Abituzumab | H:69 | 0.00 | 0.00 | 0.01 | 0.81 | 0.04 | 1.26 | 1.00 | 0.72 |
| Abituzumab | H:80 | 0.00 | 0.00 | 0.01 | 0.03 | 0.00 | 0.02 | 0.00 | 0.00 |
| Abituzumab | H:100A | 2.16 | 0.00 | 1.06 | 0.02 | 1.34 | 0.01 | 1.00 | 1.07 |
| Ofatuzumab | H:34 | 0.00 | 0.00 | 0.03 | 0.01 | 0.00 | 0.10 | 0.00 | 0.00 |
| Ofatuzumab | H:82 | 0.00 | 0.00 | 0.01 | 0.01 | 0.00 | 0.00 | 0.98 | 0.95 |
| Ofatuzumab | H:100E | 0.00 | 0.00 | 0.29 | 0.05 | 0.08 | 0.00 | 1.00 | 1.00 |
| Vesencumab | H:34 | 0.17 | 3.02 | 0.37 | 0.67 | 0.00 | 0.06 | 0.00 | 0.00 |
| Vesencumab | H:82 | 0.00 | 0.00 | 0.00 | 0.01 | 0.00 | 0.00 | 1.00 | 0.98 |
| Vesencumab | H:100B | 132.22 | 85.17 | 116.89 | 84.36 | 18.57 | 12.32 | 0.62 | 2.27 |
| Vesencumab | H:100F | 15.64 | 1.93 | 5.03 | 3.24 | 2.80 | 3.31 | 1.94 | 2.00 |

Correct predictions of positive oxidation events are highlighted in red. For the WCN-OH method, the average number of hydroxyl groups within 6 Å from the sulfur atom in the methionine side chain is reported.

Vesencumab. The method based on WCN-OH, however, correctly predicts both oxidation events for Vesencumab, H:100B and H:100F (Fig 4), in agreement with the experimental results.

Finally, we tested the hypothesis that *ab initio* calculation of the CDR-H3 loop structure might generate alternative conformations where the experimentally observed oxidized

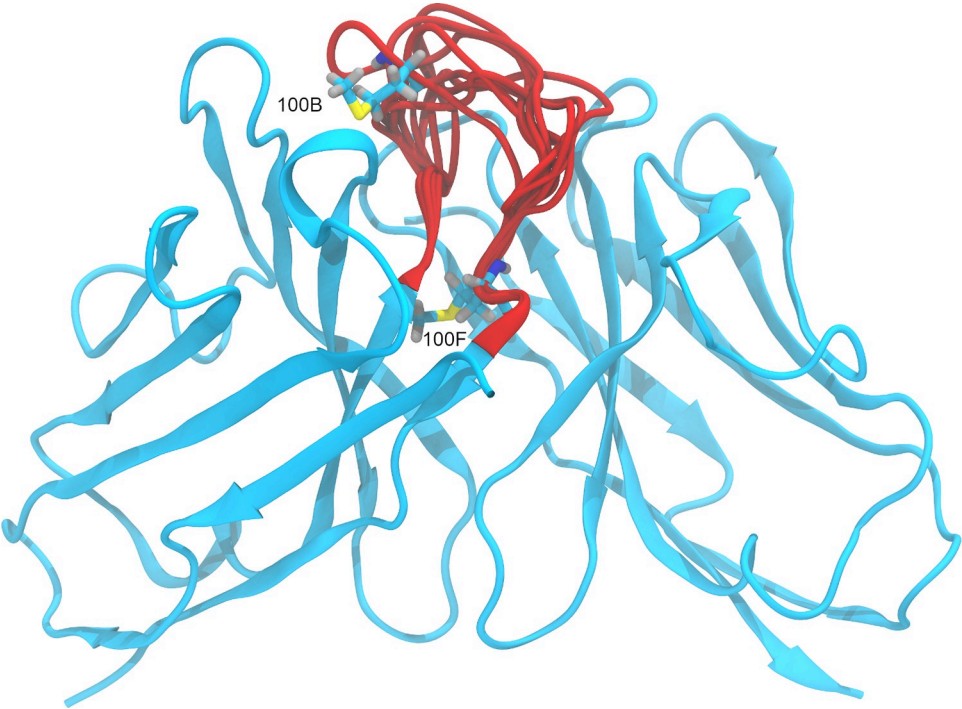

**Fig 4. Three-dimensional structure (homology model) of the Fv region of the antibody Vesencumab.** Methionines 100B and 100F, located in the H3 loop, are shown with carbon atoms colored in cyan, oxygen in red, hydrogen in gray, nitrogen in blue and sulfur in yellow. The conformations of the H3 loop obtained with Prime are represented in red.

**Table 3. Solvent-accessible surface area (SASA) of the side chain of the methionines in the Fv region of Vesencumab.**

| | sSASA ($\text{Å}^2$) | | | |
|---|---|---|---|---|
| | **H:34** | **H:82** | **H:100B** | **H:100F** |
| Crystal structure | 3.02 | 0.00 | 85.17 | 1.93 |
| model1 | 0.07 | 0.00 | 57.95 | 0.00 |
| model2 | 0.07 | 0.00 | 30.44 | 0.00 |
| model3 | 0.07 | 0.00 | 60.30 | 0.00 |
| model4 | 0.07 | 0.00 | 46.87 | 0.00 |
| model5 | 0.07 | 0.00 | 51.80 | 0.00 |
| model6 | 0.07 | 0.00 | 69.22 | 0.00 |
| model7 | 0.07 | 0.00 | 57.61 | 0.00 |
| model8 | 0.07 | 0.00 | 52.04 | 0.00 |
| model9 | 0.07 | 0.00 | 52.93 | 0.07 |
| model10 | 0.07 | 0.00 | 54.16 | 0.00 |

Values of the SASA were calculated for the crystal structure and for the 10 homology models with the H3 loop conformations calculated with Prime.

methionines are significantly solvent-exposed. For this purpose, we identified Vesencumab as the candidate for further analysis. This mAb, in fact, contains four methionines in the Fab portion of the heavy chain, two of which reside in the H3 loop (Fig 4), and two oxidation events are observed. We generated 10 models for the CDR-H3 loop using Prime [44,45] and we calculated the static SASA of the side chain for the four methionines. The results, shown in Table 3, demonstrate that as in the crystal structure three of the four methionines, except H:100B, occur with a SASA of less than 4 $\text{Å}^2$. These findings confirm that factors other than the degree of exposure to the solvent must be considered to improve the accuracy of the prediction of methionine oxidation propensity.

## Test case: Internal dataset

We demonstrated in the previous sections that the solvent accessibility of a methionine side chain, either assessed through the WCN or the SASA, is necessary but not sufficient to determine its oxidation propensity. We illustrated, in fact, that the chemical environment of the methionine moiety, in particular the presence of proximal side chains containing a hydroxyl group is a factor that affects the oxidation propensity of the residue.

To further validate the predictive methods described above, we performed forced oxidation studies on seven IgG1 antibodies and two IgG1 ADCs provided by AbbVie and we assessed the degree of oxidation of each methionine in the Fv region using peptide mapping. Consequently, we obtained experimental results that provided more detailed information for each residue (compared to the CST dataset described in the previous sections) and allowed for a better estimation of the prediction accuracy of the different methods.

As shown in Table 4 and S1 Fig, we observed that six among the twenty-six methionines considered in our experiments showed an oxidation level $\geq$ 5%, the established minimum oxidation level. For each methionine, we calculated from the three-dimensional structures and the MD trajectories the sSASA, the dSASA, the WCN and the WCN-OH parameters from the three-dimensional structure and the MD trajectory as described above for the CST antibodies dataset (S3 Table).

**Table 4. Comparison of experimental results and in silico prediction of oxidation events for the 7 mAbs and 2 ADCs.**

| | | | Position | % Ox 0.1% tBHP 24h | Predicted Ox sSASA | Predicted Ox dSASA | Predicted Ox WCN | Predicted Ox WCN-OH |
|---|---|---|---|---|---|---|---|---|
| mAb1 | Met#1 | | LFR1 | 1.7 | 0 | 0 | 0 | 0 |
| | Met#2 | | HFR2 | 3.4 | 0 | 0 | 0 | 0 |
| | Met#3 | | HFR3 | 1.4 | 0 | 0 | 0 | 0 |
| mAb2 | Met#4 | | HFR2 | 3.7 | 0 | 0 | 0 | 0 |
| | Met#5 | | HFR3 | 0.0 | 0 | 0 | 0 | 0 |
| mAb3 | Met#6 | | LFR1 | 1.3 | 0 | 0 | 0 | 0 |
| | Met#7 | | HFR2 | **34.2** | **1** | **1** | **1** | **1** |
| | Met#8 | | HFR2 | 2.1 | 0 | 0 | 0 | 0 |
| | Met#9 | | HFR3 | **9.7** | **1** | **1** | **1** | **1** |
| | Met#10 | | HFR4 | **65.6** | **1** | **1** | **1** | **1** |
| mAb4 | Met#11 | | LFR2 | 3.3 | 0 | 0 | 0 | 0 |
| | Met#12 | | HFR2 | 2.1 | 0 | 0 | 0 | 0 |
| | Met#13 | | HFR3 | 0.5 | 0 | 0 | 0 | 0 |
| | Met#14 | | HFR3 | 1.2 | 0 | 0 | 0 | 0 |
| | Met#15 | | CDRH3 | **10.0** | 0 | 0 | 0 | **1** |
| mAb5 | Met#16 | | HFR2 | 1.9 | 0 | 0 | 0 | 0 |
| | Met#17 | | HFR3 | 2.2 | 0 | 0 | 0 | 0 |
| | Met#18 | | CDRH3 | 3.1 | **1** | 0 | 0 | 0 |
| mAb6 | Met#19 | | LFR1 | 1.2 | 0 | 0 | 0 | 0 |
| | Met#20 | | LFR1 | **5.0** | **1** | **1** | **1** | **1** |
| | Met#21 | | HFR2 | 1.3 | 0 | 0 | 0 | 0 |
| mAb7 | Met#22 | | HFR2 | 2.2 | 0 | 0 | 0 | 0 |
| | Met#23 | | HFR3 | 1.1 | 0 | 0 | 0 | 0 |
| | Met#24 | | HFR4 | **39.3** | **1** | **1** | **1** | **1** |
| ADC1 | Met#25 | | HFR2 | 0.3 | 0 | 0 | 0 | 0 |
| ADC2 | Met#26 | | LFR1 | 4.6 | **1** | 0 | 0 | 0 |

Oxidation levels measured as ≥ 5% and predicted oxidation events are shown in red.

The predictions of the oxidation propensity based on the sSASA (Table 4) resulted in the incorrect assignment of two negative events and one positive event, as illustrated in the confusion matrix in Fig 5. Accordingly, this method was, for this dataset, the least accurate, with a sensitivity of 0.86±0.05 and a specificity of 0.90±0.02. The predictions based on the dSASA and the WCN provided similar results, correctly assigning all the twenty negative events and five of the six positive events. The resulting sensitivity and specificity were 0.84±0.06, 1.00±0.00 for dSASA and 0.83±0.03,1.00±0.00 for WCN, respectively. When using the WCN-OH method, all the six positive events and the twenty negative events were correctly assigned, with both the sensitivity and the specificity equal to 1.

Although all methods performed better on this set of antibodies than on the CST set, only the WCN-OH method correctly predicted all positive and negative events. The likely cause of the better performance registered by all methods was the fraction of buried methionines in the two datasets: in the CST set 91% of the methionines have a relative dSASA < 15%, whereas in the proprietary dataset (n = 9) the fraction of buried methionines is 81%. In particular, the class of methionines for which the subset of CST antibodies was selected (CDR-H3 loop) is underrepresented in the internal dataset, with only 2 elements. However, it is worth noting that for the internal dataset the sSASA, dSASA and WCN methods incorrectly identified a

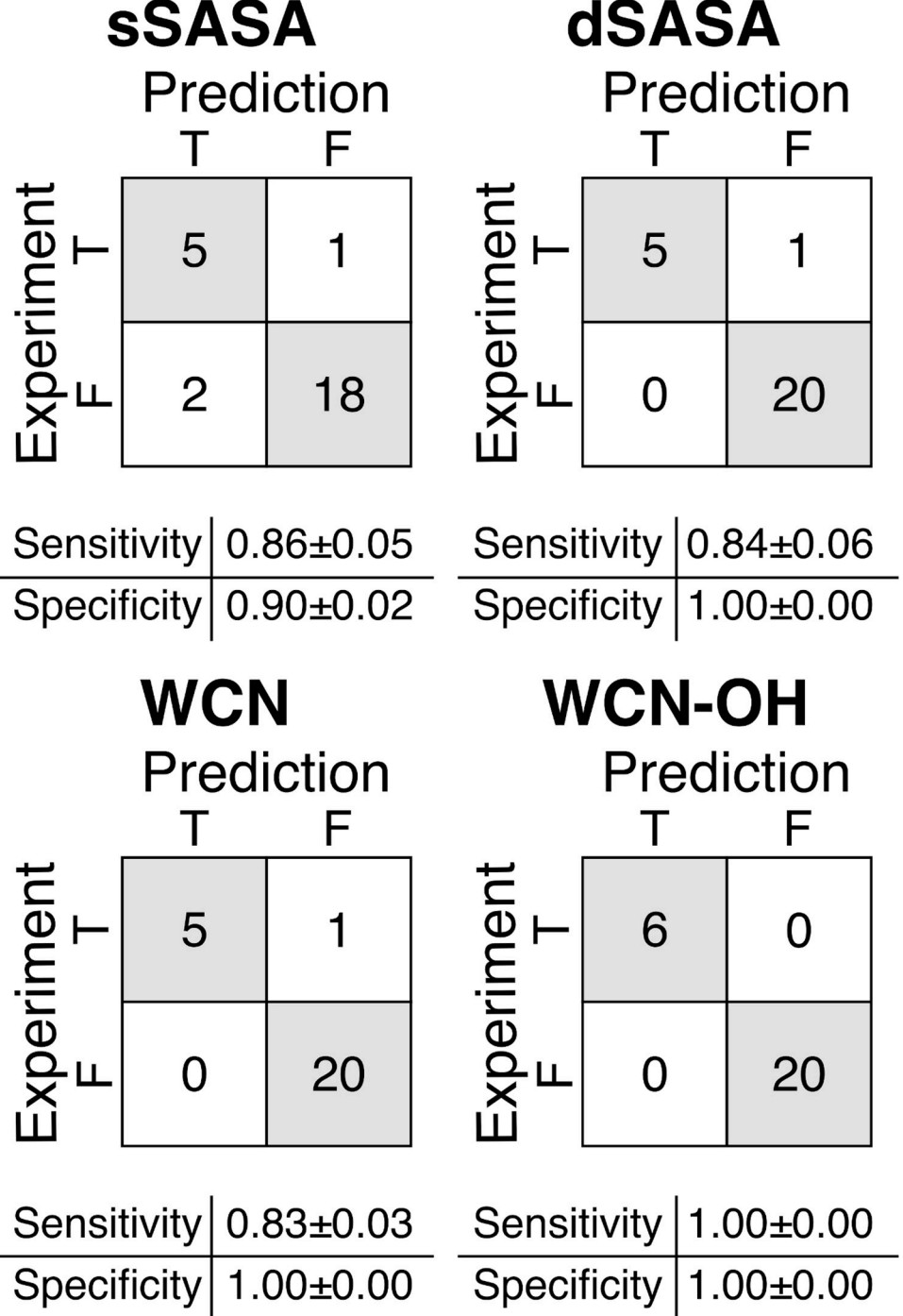

**Fig 5. Confusion matrices for the in silico prediction of the oxidation events of the 7 mAbs and 2 ADCs.**
Confusion matrices reports TP, FN on the first row and FP, TN on the second row. Sensitivity calculated as TP/(TP +FN); Specificity calculated as TN/(TN+FP). TP = True Positive, TN = True Negative, FP = False Positive, FN = False Negative. Error for the sensitivity and specificity were estimated from 50 bootstrap replicates.

methionine in the CDR-H3 loop as being non-oxidized. The chemical modification of such a residue might result in loss of antigen binding and possibly drug potency.

Furthermore, the availability in this dataset of oxidation levels for each methionine in the Fv allowed for assessement of quantitative or semiquantitative prediction based on the accessibility of the methionine side chain. We observed that WCN, sSASA and dSASA showed some degree of semiquantitative agreement with the experimental data, with values of $R^2$, correlation and Spearman correlation coefficient in the range of 0.54–0.58, 0.74–0.76 and 0.59–0.67, respectively (S1 Fig). These methods, although ineffective in predicting absolute values of oxidation rate, might still be used to rank the relative oxidation of different methionines in an antibody.

## Discussion

Methionine oxidation is a common chemical modification that occurs in therapeutic antibodies. Methionine oxidation can significantly reduce the serum circulation half-life of the antibody and, if the methionine is located in the vicinity of the CDR, it can decrease the binding affinity of the antibody with the epitope [25,34]. For this reason, identification of oxidation sites during the early stages of antibody discovery provides an engineering opportunity to remove the liable site. The information gained could also guide the subsequent formulation and production processes to reach high stability and drug potency if the candidate with an oxidation liability moves forward into development.

To date, several methods have been developed by different groups to predict *in silico* the oxidation propensity of methionine residues in proteins. Interestingly, none of them identified a consensus sequence or motif that show any correlation with the experimentally observed oxidation of the residues. Such motifs, for example, have been identified for other chemical modifications in proteins, such as deamidation [53]. Instead, accurate prediction of methionine oxidation propensity requires knowledge of the chemical environment of the sulfur atom in the methionine side chain [38]. Specifically, Chu et al. highlighted the crucial role of the water molecules surrounding the methionine side chain in stabilizing the transition state that represent the limiting step in the reaction [38]. From their seminal work several methods to assess the oxidation propensity of methionine in proteins arose, mainly based on the solvent exposure of the methionine side chain or on the water coordination number. However, as reported by Yang et al., these methods fail at predicting the oxidation propensity of at least one class of methionines in antibodies, specifically the ones situated at the end of the H3 loop in the CDR [36]. Despite these methionines having side-chain relative SASA <11%, oxidation at this site occurs frequently (Yang et al. reported oxidation in 7 out of 22 antibodies with a methionine at this position) [36]. In view of this, a deeper understanding of the mechanism underlying this reaction and a structure-based analysis of the antibodies are required to obtain reliable prediction of the oxidation liabilities.

In this work, we aimed to exploit the theoretical framework of the water coordination number to expand its prediction capability. In this context, water molecules form hydrogen bonds with hydrogen peroxide and therefore stabilize the charge separation between the two oxygens that drive the oxidation reaction. For this reason, predictive methods based on this feature associate high propensity to be oxidized with a large number of water molecules, usually 6 or more, within approximately 6 Å from the sulfur atom. An exception to this model are methionine residues at the end of the H3 loop in the CDR identified by Yang et al. [36].

In the method presented here, WCN-OH, we consider not only the water molecules surrounding the methionine side chain, but also other residues side chains containing hydroxyl groups. The rationale behind this new method lies in the observation discussed in the work of

Chu et al., that polar side chains in proximity of the methionine sulfur atom could play a role in stabilizing the transition state of the oxidation reaction by mean of hydrogen bonds with the oxidizing species [38]. We showed that, compared to other methods, WCN-OH represents a significant improvement when applied to the Yang et al. dataset of mAbs containing a partially buried methionine near the end of the H3 loop in the CDR (14 mAbs, 46 methionines). We further validated the WCN-OH method on an internal dataset of proprietary molecules (7 mAbs and 2 ADCs, 26 methionines in total). On this dataset, WCN-OH correctly predicted the oxidation propensity of 26 methionines. Interestingly, WCN-OH was the only method able to predict the oxidation of a methionine in the H3 loop of the CDR, whose modification represents a potential liability during drug development because of the risk of reduced binding affinity.

We determined the time-averaged values of WCN and SASA from 5 ns MD trajectories. Such trajectories can be collected in less than one hour on a GPU-accelerated compute node, allowing the screening of a large number of candidates during the antibody-discovery process. However, the oxidation reaction of methionine residues in antibodies occurs several orders of magnitude slower. For example, Agrawal et al. studied methionine-oxidation kinetics in the presence of 0.1% $H_2O_2$ in the Fv and Fc. The measured pseudo-first-order rate constants were 1.33 h$^{-1}$ and 0.25 h$^{-1}$, respectively [54]. Thus, extending simulations may improve performance, but any added benefit in accuracy needs to outweigh the added runtime burden. Moreover, side-chain dynamics and local conformational changes occur on a fast time scale (ps) which ns simulations accurately capture [35]. Lastly, we report that the predictions from 5 ns trajectories effectively identify oxidation-liable methionines.

The results presented here show that WCN-OH represents an improvement over current algorithm to predict methionine oxidation propensity. In particular, WCN-OH has shown to be better at predicting oxidation of methionine residues partially buried within the three-dimensional structure of the mAb. This new method not only improves prediction accuracy, but also provides additional atomic-level insight of the methionine oxidation mechanism. We anticipate that prospective application of the WCH-OH method on more extensive datasets will further validate its improved accuracy and applicability to antibody-drug development.

## Supporting information

**S1 Fig. Comparison of experimental results and in silico prediction of oxidation events for the 7 proprietary mAbs and 2 ADCs.** Experimental oxidation level and calculated descriptor for each methionine in consideration are shown in the plot. Vertical red lines indicate oxidation levels measured as $\geq$ 5%, horizontal red lines indicate threshold values for the different descriptors.
(TIF)

**S1 Table. Comparison of experimental results and in silico prediction of oxidation events for 4 proprietary molecules.** Oxidation levels measured as $\geq$ 5% are shown in red.
(DOCX)

**S2 Table. Methionine side chain accessibility parameters calculated from MD trajectories for the CST antibodies.**
(DOCX)

**S3 Table. Methionine side chain accessibility parameters calculated from MD trajectories for the AbbVie antibodies.**
(DOCX)

**S1 File. Details for the preparation and analysis of ADC1-2 and for replicates of mAb2, mAb4, mAb7.**
(DOCX)

## Acknowledgments

The authors would like to thank Yao Fan, Ph.D., of AbbVie, for helpful discussion.

## Author Contributions

**Conceptualization:** Davide Tavella, Kai Zhu, Jianwen Xu, Eliud O. Oloo, Christopher Negron, Peter M. Ihnat.

**Data curation:** Davide Tavella, David R. Ouellette, Raffaella Garofalo.

**Formal analysis:** Davide Tavella, David R. Ouellette, Raffaella Garofalo, Christopher Negron.

**Funding acquisition:** Davide Tavella.

**Investigation:** Davide Tavella, David R. Ouellette, Raffaella Garofalo, Kai Zhu, Jianwen Xu, Christopher Negron, Peter M. Ihnat.

**Methodology:** Davide Tavella, David R. Ouellette, Raffaella Garofalo, Kai Zhu, Eliud O. Oloo, Christopher Negron.

**Project administration:** Davide Tavella.

**Resources:** Davide Tavella, Kai Zhu, Eliud O. Oloo.

**Software:** Davide Tavella, Kai Zhu.

**Supervision:** Davide Tavella.

**Validation:** Davide Tavella.

**Visualization:** Davide Tavella.

**Writing – original draft:** Davide Tavella, David R. Ouellette, Raffaella Garofalo, Kai Zhu, Jianwen Xu, Eliud O. Oloo, Christopher Negron, Peter M. Ihnat.

**Writing – review & editing:** Davide Tavella, David R. Ouellette, Raffaella Garofalo, Kai Zhu, Jianwen Xu, Eliud O. Oloo, Christopher Negron, Peter M. Ihnat.

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
