## [Decision Letter · Decision Letter 0]

31 Aug 2022

PONE-D-22-16362A novel method for in silico assessment of Methionine oxidation risk in monoclonal antibodies: improvement over the 2-shell model.PLOS ONE

Dear Dr. Tavella,

Thank you for submitting your manuscript to PLOS ONE. After careful consideration, we feel that it has merit but does not fully meet PLOS ONE’s publication criteria as it currently stands. Therefore, we invite you to submit a revised version of the manuscript that addresses the points raised during the review process.

We look forward to receiving your revised manuscript.

Kind regards,

Ioscani Jimenez del Val, Ph.D.

Academic Editor

PLOS ONE

Journal Requirements:

“The design, study conduct, and financial support for this research were provided by AbbVie. AbbVie participated in the interpretation of data, review, and approval of the publication. Davide Tavella, Christopher Negron, David R. Ouellette, Raffaella Garofalo and Jianwen Xu are employees of AbbVie and own AbbVie stock. Peter M. Ihnat was an employee of AbbVie at the time of this study. . Desmond, Prime, Maestro and BioLuminate are products sold by Schrodinger, Inc. Eliud O. Oloo and Kai Zhu performed this research as employees of Schrödinger, Inc.”           

Reviewers' comments:

Reviewer's Responses to Questions

**Comments to the Author**

1. Is the manuscript technically sound, and do the data support the conclusions?

Reviewer #1: Yes

Reviewer #2: Yes

2. Has the statistical analysis been performed appropriately and rigorously? 

Reviewer #1: Yes

Reviewer #2: Yes

3. Have the authors made all data underlying the findings in their manuscript fully available?

Reviewer #1: Yes

Reviewer #2: No

4. Is the manuscript presented in an intelligible fashion and written in standard English?

Reviewer #1: Yes

Reviewer #2: Yes

5. Review Comments to the Author

Reviewer #1: The solvent-accessible surface area (SASA) of the methionine side chain from a predicted antibody structure is a commonly used parameter to predict

oxidation propensity of methionine residues in mAbs. Here Tavella and report an improved algorithm to predict methionine oxidation propensity (eg for partially buried Met). The algorithm considers not only the water molecules surrounding the methionine side chain, but also other residues side chains containing hydroxyl groups. The data are convicing (prediction vs forced degraded mAbs and ADCs), the manuscript is well written and of interest for the increasing stakeholders in antibody-based drugs. Altogether, the paper deserves to be published in PLOS One.

Reviewer #2: The article on Met oxidation of MAbs is lucid and interesting. I endorse its ultimate publication as a good contribution to our understanding adventitious met oxidation, both in therapeutic molecules and in the cell.

While I understand the focus on go/no go criteria to drive pharma decisions, well supported with the confusion matrices, I'd prefer some explicit statistical predictions of "oxidation riak" shall we say. Reporting a binary cutoff at a 15%? threshold is nice but why is that the right number? Cna the authors share mnre results from their calculations or present them in an alternate way, even as supporting material?

A second point concerns MS data, will the new generated be made available in PRIDE or other database? This is important to the study's impact.

Lastly I'm wondering about timescales. For calculating time averaged surface excursions of partially buried Met, and thinking about chemical reaction dynamics, what % of Met are available for chemical reaction at any time? Does this fit with the concentrations and reactivity of the oxidation agents used?

6. PLOS authors have the option to publish the peer review history of their article (what does this mean?). If published, this will include your full peer review and any attached files.

Reviewer #1: No

Reviewer #2: No

---

## [Author Response · Author response to Decision Letter 0]

19 Nov 2022

Responses to editor’s comments:

We have corrected the format of the authors’ affiliation as per template instructions. We have changed the style and font size of sections’ and subsections’ titles. We have moved the captions of all figures at the end of the sections where they have been firstly referred to, and we have adjusted the captions’ style. We have moved all tables at the end of the sections where they have been firstly referred to. We have attached to this submission all the figures as separated files named accordingly to the style templates. Figures have been assessed by PACE. We hope that it now fits the style requirements, as described in the referred templates.

We have changed the “Disclosure” section in order to include the sources of funding for the

work included in this submission. No grants were awarded to any authors. All authors participated to this work as employees of AbbVie or Schrödinger, and AbbVie sponsored and funded the study.

“The design, study conduct, and financial support for this research were provided by AbbVie. AbbVie participated in the interpretation of data, review, and approval of the publication. Davide Tavella, Christopher Negron, David R. Ouellette, Raffaella Garofalo and Jianwen Xu are employees of AbbVie and own AbbVie stock. Peter M. Ihnat was an employee of AbbVie at the time of this study. . Desmond, Prime, Maestro and BioLuminate are products sold by Schrodinger, Inc. Eliud O. Oloo and Kai Zhu performed this research as employees of Schrödinger, Inc.” 

We have changed the “Disclosure” section as follow:

AbbVie sponsored and funded the study, contributed to the design, participated in the collection, analysis, and interpretation of data, and in writing, reviewing, and approval of the final publication. Davide Tavella, Christopher Negron, David R. Ouellette, Raffaella Garofalo and Jianwen Xu are employees of AbbVie and may own AbbVie stock. Peter M. Ihnat was an employee of AbbVie at the time of this study. Desmond, Prime, Maestro and BioLuminate are products sold by Schrödinger, Inc. Schrödinger, Inc. provided support in the form of salary for authors Eliud O. Oloo and Kai Zhu but did not have any additional role in the study design, data collection and analysis, decision to publish or preparation of the manuscript. The specific roles of these authors are articulated in the “author contributions”.

The roles of the two funders, AbbVie and Schrödinger, have been explicitly stated as per provided instructions.

In this work, we have used two experimental datasets to validate the in silico predictions of methionine oxidation propensities and to reach the conclusions drawn in the manuscript.

The first dataset, referred to in the manuscript as “clinical stage therapeutic (CST) antibodies dataset”, was derived from existing data, which are openly available at https://doi.org/10.1080/19420862.2017.1290753. This is the largest dataset used in this work, counting 14 antibodies and a total of 46 methionines.

The second dataset, referred to in the manuscript as “internal dataset”, is derived from antibodies and ADCs currently in development at AbbVie. Therefore, due to its proprietary nature, supporting data cannot be made openly available. This is the smallest dataset used in this work, counting 7 antibodies and 2 ADCs for a total of 26 methionines.

We want to highlight that the conclusions drawn from both datasets are in total agreement and the “internal dataset” serves as a further validation of the methods described in the manuscript.

Moreover, in the supporting information S2 Table and S3 Table, we have disclosed more details regarding the calculations derived from the molecular dynamics simulations for both datasets.

We have added captions for Supporting Information Tables and Figures at the end of the manuscript.

We have revised the reference list to match the style and format of the journal. We have included a new reference (reference 49 in the final version). No retracted articles are referenced in the manuscript.

Responses to reviewer #2:

Reviewer #2: The article on Met oxidation of MAbs is lucid and interesting. I endorse its ultimate publication as a good contribution to our understanding adventitious met oxidation, both in therapeutic molecules and in the cell.

While I understand the focus on go/no go criteria to drive pharma decisions, well supported with the confusion matrices, I'd prefer some explicit statistical predictions of "oxidation riak" shall we say. Reporting a binary cutoff at a 15%? threshold is nice but why is that the right number? Cna the authors share mnre results from their calculations or present them in an alternate way, even as supporting material?

We agree with the reviewer’s points, and thus we included a statistical analysis of the linear correlation between experimentally measured oxidation levels of methionines and parameters calculated from the molecular dynamics simulations, specifically water coordination number and SASA. These results are reported in the modified S1 Fig and discussed in the result section at lines 443-450. The statistical analysis includes R-squared, correlation coefficient and Spearman correlation coefficient.

The choice of 15% threshold is now discussed in the result section at line 281-282.

We added 2 tables, S2 Table and S3 Table in supporting information, with details of the calculations from the molecular dynamics simulations.

A second point concerns MS data, will the new generated be made available in PRIDE or other database? This is important to the study's impact.

As discussed in the response to Editor’s comment #4, the dataset indicated as “internal dataset” is derived from antibodies and ADCs currently in development at AbbVie. Therefore, due to its proprietary nature, supporting data cannot be made openly available.

The “clinical stage therapeutic (CST) antibodies dataset” was derived from existing data, which are openly available at https://doi.org/10.1080/19420862.2017.1290753.

Lastly I'm wondering about timescales. For calculating time averaged surface excursions of partially buried Met, and thinking about chemical reaction dynamics, what % of Met are available for chemical reaction at any time? Does this fit with the concentrations and reactivity of the oxidation agents used?

We have added a paragraph that address the different time scales involved in the molecular dynamics simulations and the kinetics of the oxidation reaction in the Discussion section at lines 514-525.

---

## [Decision Letter · Decision Letter 1]

13 Dec 2022

A novel method for in silico assessment of Methionine oxidation risk in monoclonal antibodies: improvement over the 2-shell model.

PONE-D-22-16362R1

Dear Dr. Tavella,

We’re pleased to inform you that your manuscript has been judged scientifically suitable for publication and will be formally accepted for publication once it meets all outstanding technical requirements.

Kind regards,

Ioscani Jimenez del Val, Ph.D.

Academic Editor

PLOS ONE

Additional Editor Comments (optional):

Thank you for addressing all reviewer comments/suggestions.

Reviewers' comments:

Reviewer's Responses to Questions

**Comments to the Author**

1. If the authors have adequately addressed your comments raised in a previous round of review and you feel that this manuscript is now acceptable for publication, you may indicate that here to bypass the “Comments to the Author” section, enter your conflict of interest statement in the “Confidential to Editor” section, and submit your "Accept" recommendation.

Reviewer #1: All comments have been addressed

2. Is the manuscript technically sound, and do the data support the conclusions?

Reviewer #1: Yes

3. Has the statistical analysis been performed appropriately and rigorously? 

Reviewer #1: Yes

4. Have the authors made all data underlying the findings in their manuscript fully available?

Reviewer #1: Yes

5. Is the manuscript presented in an intelligible fashion and written in standard English?

Reviewer #1: Yes

6. Review Comments to the Author

Reviewer #1: (No Response)

7. PLOS authors have the option to publish the peer review history of their article (what does this mean?). If published, this will include your full peer review and any attached files.

Reviewer #1: **Yes: **Alain Beck

---

## [Editor Report · Acceptance letter]

19 Dec 2022

PONE-D-22-16362R1 

A novel method for *in silico* assessment of Methionine oxidation risk in monoclonal antibodies: improvement over the 2-shell model. 

Dear Dr. Tavella:

I'm pleased to inform you that your manuscript has been deemed suitable for publication in PLOS ONE. Congratulations! Your manuscript is now with our production department. 

Kind regards, 

on behalf of

Dr. Ioscani Jimenez del Val 

Academic Editor

PLOS ONE